# Differences in Agonistic Behavior and Energy Metabolism between Male and Female Swimming Crab *Portunus trituberculatus* Based on the Analysis of Boldness

**DOI:** 10.3390/ani12182363

**Published:** 2022-09-10

**Authors:** Xianpeng Su, Boshan Zhu, Ziwen Ren, Fang Wang

**Affiliations:** 1The Key Laboratory of Mariculture, Ministry of Education, Ocean University of China, 5 Yushan Road, Qingdao 266003, China; 2Shandong Yellow River Delta Marine Technology Co., Ltd., Longtai Road, Xinhu Town, Hekou, Dongying 257000, China; 3Function Laboratory for Marine Fisheries Science and Food Production Processes, Qingdao National Laboratory for Marine Science and Technology, No. 1 Wenhai Road, Aoshanwei Town, Jimo, Qingdao 266237, China

**Keywords:** *Portunus trituberculatus*, crab behavior, machine learning, fighting intensity, crab personality

## Abstract

**Simple Summary:**

Boldness is a widely studied personality trait that is positively related to an individual’s willingness to fight. To explore the agonistic behavior between female and male crabs with different degrees of boldness, we quantified the boldness of females and males using a behavior observation system and used the experimentally determined boldness and classification. The results showed that boldness affected the agonistic behavior between female and male swimming crabs, the fighting willingness and capacity of male crabs is higher than that of female individuals and is higher in bold crabs than in shy individuals. Energy reserves and metabolic rates may be one of the factors affecting the agonistic behavior in crabs and that the agonistic behavior resulted in significant changes in key energy metabolites.

**Abstract:**

Individual differences in metabolism and agonistic behavior have been a key research area in evolution and ecology recently. In this study, we investigated the boldness of swimming crabs *Portunus trituberculatus* and explored the agonistic behavior between female and male crabs, specifically examining competitions between bold females vs. bold males (BF–BM), bold females vs. shy males (BF–SM), shy females vs. shy males (SF–SM), and shy females vs. bold males (SF_BM) and its relationship with energy metabolism. The main results revealed the following: There was no significant difference in boldness between females and males, while there were more bold individuals than shy in both females and males. Bold individuals initiated significantly more fights than shy individuals, and male initiators won significantly more fights than female initiators. The duration and intensity of fight between bold individuals was significantly higher than fights between shy individuals. For males, the concentration of glucose in the hemolymph was significantly higher in shy crabs than bold crabs, while there was no significant difference between shy and bold individuals in females. After fighting, the concentration of glycogen in claws was lower than that before fighting, and the concentrations of glucose and lactate in hemolymph were significantly higher after fighting than before. We found that the fighting willingness and ability were higher in male crabs than females and higher in bold crabs than shy. Fighting ability varied between sexes and was influenced by boldness and energy state.

## 1. Introduction

Agonistic behavior, an important form of internal competition, is a fundamental aspect of ecological and behavioral research of crustaceans [1]. Aggressions are often resolved by behavior and aggressivity evaluation of individuals, which advertise fighting ability or resource holding potential (RHP) [2,3]. Asymmetries in RHP result from physical and physiological difference between individuals, such as size [4], sex [5], energetic status [6,7], and hormone levels [4], all of which directly indicate the likelihood of an individual winning a fight. Briffa and Elwood [8] suggested that the essence of fight is the competition of total energy reserve of individuals where the eventual victor commits the greatest amount of energy and consumes the most energy substrate such as glucose and glycogen.

During fighting, oxygen consumption of crustaceans is directly utilized in energy metabolism [9,10]. The consumption and reserve of glycogen and glucose and the accumulation of lactate reflects the energy state of individuals and is related to the result of fighting [8,11]. For example, hermit crabs *Pagurus bernhardus*, whose hemolymph glucose concentration is high, have higher aggressiveness and are more likely to succeed in competing for resources [7]. At the beginning stage of fighting, intense agonistic activity is fueled partly by anaerobic metabolism [12,13]. However, as a consumed activity, the aggression and fight intensity may be restricted because of the limited energy supply [8,14,15].

Stable and consistent behaviors across multiple contexts [16] and these consistent behavioral differences among individuals were named as animal personality [17,18], which affects the fitness correlates, such as growth, reproduction, survival, and population dispersion [19,20]. Because of its link to fitness-related traits, animal personality also has important ecological and evolutionary implications [21]. It has also become evident that personalities may be related to consistent differences in physiological states and that personality could potentially reveal different relationships between RHP and different strategies during and following contests [22]. Among the normal personality category, boldness, positively relating to individual’s agonistic behavior [23], is a measurement of the reaction in a dangerous situation, which is evaluated according to the shy–bold axis [24,25]. Animals are assigned to bold or shy types according to their assessment in the bold–shy dichotomy [24]. In relative research in crustaceans, it has been confirmed that bold individuals are more likely to win in the competition, while the shy have a competitive advantage under particular circumstance [25,26]. For example, bold fiddler crabs *Uca mjoebergi* initiate most fights in territorial conflicts [27], while shy hermit crabs *P. bernhardus* win more contests defending their shells [22]. Sex, as a defining intrinsic factor, is not only related to personality traits, but also affects the agonistic behavior of individuals. In the hermit crab *P. bernhardus*, males fight more intensely, and male attackers succeed more often than females who attack [5]. However, these differences in agonistic performance cannot be attributed exclusively to differences in perceived fitness traits or differences in personality traits between the sexes.

The swimming crab *Portunus trituberculatus* is an aggressive crustacean in the shelf of the West Pacific Ocean. As an important aquacultural and ecological species, intra-species fighting and cannibalism occurs when they are competing for access to resources [28,29], which is related to boldness [25]. The previous studies have focused on male crabs, but it is still not clear if there is a difference in agonistic behavior and boldness between male and female crabs [20]. We hypothesized that fighting willingness, ability, and energy state of crabs with different gender are influenced by boldness. To test this hypothesis, we quantified the boldness of females and males using a behavior observation system and used the experimentally determined boldness and classification. We then analyzed the agonistic behavior, concentrations of energy reserves, and oxygen consumption rates. In addition, the influence of an individual’s physiological state on the fight outcome was explored. In culture, this study provided a reference for the decline in cannibalism and improvement in the survival rate of swimming crab. These data will help strengthen our mechanistic understanding of the causal links between physiology and behavior in various scenarios.

## 2. Materials and Methods

### 2.1. Animal Collection and Maintenance

The study was completed during August 2019 at the Key Laboratory of Mariculture, Ministry of Education, Ocean University of China, Qingdao, China. Both male (carapace width: 92.4 ± 6.3 mm) and female (carapace width: 91.3 ± 6.4 mm) swimming crabs (*P. trituberculatus*) were collected from the aquaculture facility in Jiaonan, Qingdao. Each crab was cultured in the aquarium (40.5 L, 45 × 30 × 30 cm) that was continuously aerated for acclimatization separately. During acclimatization, the photoperiod was 12 h: 12 h (Light: Dark), the temperature of seawater, which was exchanged half daily, was maintained at 24 ± 0.5 °C, and the salinity was 30%. Crabs were fed at 08:00 every morning with adequate Manila clams *Ruditapes philippinarum*, and shells and residual clams were removed 4 h after feeding.

### 2.2. Estimate of Boldness

In order to record and analyze the behavior of crabs to measure boldness, we built a personality recording system (Figure 1a). The system consisted of a cylindrical experimental aquarium (diameter = 60 cm, height = 80 cm), an infrared camera (HIKVISION, DS-IPC-B12HV2-IA, Hangzhou, China), a monitor (PHILLIP, 233i, Dongguan, China), and a box shelter (30 × 20 × 20 cm) with a trapdoor (20 × 20 cm) (Figure 1a) [30]. The infrared camera was installed 0.7 m above the experimental aquaria. Depth of water in the aquaria was maintained at 40 cm. In order to complete the experiment, 16 identical video capture systems were set.

In the experiment, only healthy (all appendages intact, normal ingestion) female (carapace width: 90.6 ± 5.4 mm, *n* = 48) and male (carapace width: = 89.1 ± 3.9 mm, *n* = 48) individuals in intermolt stage with intact appendages were used only once. To measure the boldness of crabs, individuals were left in the shelter for 10 min to acclimatize. Then the door was opened gently to allow the crabs to access the shelter optionally, and their behavior was recorded using a video camera. The recording lasted for 24 h. After the recording, the crabs were put back into the initial aquaria for recovery. Boldness was estimated as the proportion of time that crabs were out of the shelter during the recording time (24 h) [20,25]. After that, the personality analysis method based on machine learning (PAML) was used for the bold–shy classification in male and female crabs [24]. Therefore, there were four types of individuals in this experiment: bold females (BF), shy females (SF), bold males (BM), and shy males (SM).

### 2.3. Estimate of Agonistic Behavior

We built an agonistic behavior recording system in the laboratory (Figure 1b). The components were basically the same as the personality recording system. The difference was that the shelter was replaced by a separator board (60 × 50 cm). In order to reduce the error caused by different experimental time, 16 identical observation systems were set and used synchronously for collecting agonistic behavior data. In the present study, to explore the agonistic behavior between female and male crabs with different boldness, all four treatment groups of intersexual fighting were conducted, namely, bold females vs. bold males (BF–BM), bold females vs. shy males (BF–SM), shy females vs. shy males (SF–SM), and shy females vs. bold males (SF–BM). Clams were withdrawn 24 h before the experiment. Two crabs with similar carapace widths from different boldness/gender categories were selected to complete a fight. The two selected crabs were separated by a board and were allowed an adaptation time of 10 min [30]. Crabs were marked with a white dot using acrylic paint on the carapace for distinction and observation. In order to promote the intensity of fighting, after seawater added, 5 ml food extract (clam mantle homogenized and diluted with seawater) was injected into the arena and distributed evenly [10]. After the settling time (10 min), the separating board was raised gently, and the behavior of the crabs was recorded by the infrared cameras for 45 min. A total of 24 fights were conducted in each fight treatment. The behavioral data were stored using a mobile hard disk (Seagate, STJL2000400) and then analyzed. After all fighting completed, the duration of the fight in each treatment group was recorded, and the intensity of the fighting was assessed. The duration time was calculated from the moment one crab initialized the attack until the two crabs stopped bodily contact [14]. The intensity of fighting was determined as strong, moderate, weak, and very weak (Table 1) [14,31].

The agonistic behaviors of each crab were counted and analyzed according to Sneddon et al. (Table 2) [14]. The frequency of behavior refers to the total number of agonistic behavior performed by crabs. The initiator, which was defined as the crab moving toward and contacting its opponent first, and the winner, which won more frequently during shooting, in each fighting were recorded. The winner of a fight was the crab that forced its opponent to retreat repeatedly or climbed on the carapace of another contestant successfully [14]. We also recorded the number of winning bouts and losing bouts of the initiator (Noted as init/win and init/loss, respectively).

### 2.4. Measurement of the Metabolism Parameters

#### 2.4.1. Oxygen Consumption Rate

Metabolic rates are typically estimated using oxygen consumption rates (OCR) [32,33]. In the present study, we measured OCR of the four types of individuals by respirometry [33]: bold females, shy females, bold males, and shy males. In order to facilitate the transfer of crabs, the respiratory bottle was replaced with a round beaker (16 × 25 cm, diameter × height; the volume of 5000 mL). Three blank OCR controls (without crabs) were also measured. After acclimation, the gate valves were shut down, and the experiment lasted 45 min. Dissolved oxygen of the water samples was measured with YSI Pro20 (Teflon; Pro20; Yellow Springs, OH, USA). OCR (mg g^−1^ h^−1^) were calculated with
OCR = (*C*_0_−*C_t_*) × *V* × *W^−1^* × *T^−1^*
where *C*_0_ (mg mL^−1^) is the final dissolved oxygen concentration of the blank bottle; *C_t_* (mg mL^−1^) is the final dissolved oxygen concentration of the experimental bottle; *V* (mL) is the respiratory bottle cubage; *W* (g) is the wet bodyweight of swimming crabs, and *T* (h) is experimental time [33].

#### 2.4.2. Energy Metabolism Parameters

We measured the concentration of glycogen and glucose in the four types before fighting [30]. Eight pairs of bold females vs. bold males fights were selected to determine the glycogen, glucose, and lactate concentration. After fighting, crabs were moved from the experimental aquarium with minimal disturbance and anesthetized using ice [20], and a hypodermic needle with a 1 mL syringe was used to take 2 mL hemolymph samples from the base of the last pereopod [25]. After hemolymph was collected, the carapace and claw were removed to kill the crab quickly [11]. Muscle in the claw was then scraped out with tweezers, frozen in microtubes with liquid nitrogen, and stored at −80 °C for further analyses. The hemolymph samples were centrifuged for 10 min at 6000 rpm at 4 °C; the supernatants were separated and collected and frozen at −80 °C [11]. All steps were completed on ice.

Using the protocol of commercial assay kits, the anthrone–sulfuric acid colorimetric method was used to measure the concentration of glycogen in the claw tissue (mg g^−1^) using spectrophotometry at 620 nm. The glucose oxidase–peroxidase method was used to measure the concentration of glucose in the hemolymph (mmol L^−1^) using spectrophotometry at 505 nm. The NBT colorimetric method was used to measure the concentration of lactate in the hemolymph (mmol L^−1^). The result was measured using spectrophotometry at 530 nm. An automatic microplate reader (Synergy^2^, BioTek, Winooski, VT, USA) was used to read the OD values [11,30].

### 2.5. Statistical Analysis

SPSS 25.0 was used for all data analysis. All data were expressed as mean ± SD. Differences in intensity were compared using the Kruskal–Wallis test. Paired *t*-test was used to compare the difference of the frequencies of agonistic behavior in the matched crabs. A chi-square test was used to compare the numbers of bold and shy individuals in males and females, as well as the difference of the winning percentage of the initiator in different types. The duration time and the frequency of agonistic behavior among different groups was analyzed by one-way ANOVA. The Duncan post-hoc comparison test was used to evaluate differences. The concentration of glycogen, glucose, and OCR was analyzed using two-way ANOVAs. The Duncan post-hoc comparison test was used to evaluate differences. In the BF–BM group, the concentration of glycogen, glucose, and lactate was compared by two-way ANOVAs. When analyzing, ANOVAs with sex and boldness were set as the fixed factors. Prior to analysis, the Shapiro–Wilk was used to check the normality, and the homogeneity of variance was checked by a Levene’s test; *p* < 0.05 was treated as the significant difference in all analyses.

## 3. Results

### 3.1. Boldness of Female and Male Swimming Crabs

The results showed that there was no significant difference in boldness between female and male swimming crabs (*P. trituberculatus*) (*t*-test, *p* = 0.579). The boldness (proportion of time outside shelter) of bold individuals was mainly concentrated between 0.5–1.0, while shy individuals were mainly concentrated between 0–0.5 (Figure 2). The number of bold individuals was significantly higher than that of shy individuals in both females and males (Chi-square test, χ^2^_female_ = 21.33, *p* < 0.01; χ^2^_male_ = 14.08, *p* < 0.01).

### 3.2. Duration Time and Intensity of Fighting

The duration time of fighting varied significantly among different treatment groups (one-way ANOVA, MS = 655.87, *F* = 9.78, *p* < 0.01). The duration time was longer in the BF–BM group, which was significantly higher than that in the SF–SM group (*p* < 0.05), and the duration time in both the BF–SM and SF–BM groups was significantly higher than that in the SF–SM group (*p* < 0.05) (Figure 3a). The intensity of fighting varied significantly among different treatment groups (Kruskal–Wallis, H = 22.28, *df* = 3, *p* < 0.01). Fighting intensity in the SF–SM group was significantly lower than that in other groups (*p* < 0.05), and the intensity of fighting in the SF–SM group was mainly assessed as very weak and weak, while in the BF–BM group it was mainly evaluated as weak and moderate (Figure 3b). The SF–SM group had the highest percentage of very weak, the BF–BM group had the highest percentage of weak, the BF–SM group had the highest percentage of moderate, and the SF–BM group had the highest percentage of strong (Figure 3b).

### 3.3. Agonistic Behavior

In the SF–BM group, the number of fights initiated by male crabs was significantly higher than female (Table 3, chi-square test, χ^2^_SF-BM_ = 4.17, *p* = 0.04). Moreover, in the BF–SM and SF–BM groups, the number of wins by male crabs was significantly higher than with female crabs as the initiators (Table 3, χ^2^_BF–SM_ = 4.46, *p* = 0.03; χ^2^_SF–BM_ = 5.33, *p* = 0.02). In the BF–BM and SF–SM groups, there were no significant differences in the number of fights initiated by individuals and the number of wins as the initiators between males and females (Table 3, *p* > 0.05).

The frequency of agonistic behavior varied among different groups (Figure 4). In the SF–SM group, the total number of “move to” and “contact behaviors” were significantly lower than that of BF–BM groups (Figure 4a,d). In the SF–BM group, the total number of “move away” was significantly higher than that of the BF–SM and SF–SM groups (Figure 4b). In the SF–BM group, the total number of “cheliped display” of crabs was significantly higher than that of the other (Figure 4c). In the BF–BM and SF–BM groups, the total number of “move to” by males was significantly higher than that by the females (paired *t* test, BF–BM, *t* = −7.63, *p* < 0.01; SF–BM, *t* = −5.96, *p* < 0.01). However, in the SF–SM groups, the times of “move to” by females were significantly higher than those by the males (Figure 4a, paired *t* test, *t* = 2.11, *p* = 0.04). In the BF–BM, BF–SM, and SF–BM groups, the times of “move away” by females were significantly higher than those of the males (paired *t* test, BF–BM, *t* = 4.36, *p* < 0.01; BF–SM, *t* = 5.02, *p* < 0.01; SF–BM, *t* = 4.43, *p* < 0.01). However, there were no significant differences in the times of “move away” in the SF–SM group between two matched crabs (paired *t* test, *t* = −0.58, *p* = 0.56) (Figure 4b). In the BF–BM, BF–SM, and SF–BM groups, the times of “cheliped display” of males were significantly higher than those of the females (paired *t* test, BF–BM, *t* = −4.23, *p* < 0.01; BF–SM, *t* = −2.41, *p* = 0.03; SF–BM, *t* = −3.82, *p* = 0.01), but there were no significant differences in the times of “cheliped display” in the SF–SM group between two matched crabs (Figure 4c, paired *t* test, *t* = 1.19, *p* = 0.24). The times of contact behaviors of the BF–SM, BF–BM, and SF–BM groups were significantly higher than the SF–SM group. In the BF–BM and SF–BM group, the times of contact behaviors of males were significantly higher than those of the females (Figure 4d, paired *t* test, BF–BM: *t* = 3.44, *p* < 0.01; SF–BM: *t* = 4.81, *p* < 0.01).

### 3.4. Energy Metabolism Parameters

#### 3.4.1. Energy Metabolism between Different Types of Individuals

There was no significant difference in the concentration of glycogen in the claw of the four types of crabs (Table 4; Figure 5a). The concentration of glucose in the claw of swimming crabs was significantly affected by Boldness and sex (Table 4, MS = 1.697, *F* = 4.508, *p* = 0.045), the glucose concentration in the hemolymph of bold females was significantly higher than that of bold males (*p* < 0.001), while there was no significant difference in glucose levels between shy female and shy male individuals (Figure 5b). The glucose concentration in the hemolymph of shy males was significantly higher than that of bold males (*p* < 0.05), while there was no significant difference between bold female and shy female individuals (Figure 5b). The OCR of males was significantly higher than that of female individuals (Figure 5c).

#### 3.4.2. Energy Metabolism before and after Fighting

The concentration of glycogen in the claw of swimming crabs was significantly affected by fighting and sex (Table 5, Fighting, MS = 0.509, *F* = 8.357, *p* = 0.005; Sex, MS = 0.376, *F* = 6.175, *p* = 0.015). The concentration of glycogen in claws before fighting was significantly higher than after (Figure 6a). The concentration of glucose in the hemolymph of swimming crabs was significantly affected by fighting (Table 5, MS = 5.968, *F* = 16.590, *p* = 0.001). Glucose concentration in the hemolymph before fighting was significantly lower than after fighting for males, while there was no significant difference in female individuals before and after fighting (Figure 6b). Lactate concentration in the hemolymph of swimming crabs was significantly affected by fighting (Table 5, MS = 0.002, *F* = 4.041, *p* = 0.047). The concentration of lactate in the hemolymph before fighting was significantly lower than after fighting for both female and male individuals (Figure 6c).

## 4. Discussion

Performance of individuals during fighting is influenced by the capacity and willingness to defeat the opponent [25]. The willingness to fight is known to be determined by both intrinsic (e.g., sex and personality traits) and extrinsic (e.g., fighting experience and resource value) factors [34,35]. Boldness is a widely studied personality trait that is positively related to an individual’s willingness to fight [36]. In this study, consistent with the hypothesis, the longer duration time and higher intensity of a fight involving bold individuals showed that bold crabs have a higher willingness to compete during a fight than shy individuals, which was similar to the fiddler crabs *U. mjoebergi* [27] and our previous observations of swimming crabs [30]. As an important intrinsic factor, sex is not only related to personality traits, but also affects the agonistic behavior of individuals. For example, male cichlids *Astatotilapia burtoni* were more willing to fight than females and were also more likely to win fights and gain the dominant positions in the population [37]. In our experiments, more fights were initiated by male crabs than by female crabs, and the male crabs initiating fights were more likely to win. These results indicated that males were more willing and able to fight than females. The performance in fighting also reflects the willingness and capacity of individuals to compete, and exhibiting agonistic behaviors frequently is a logical indicator of the dominance during fights [37,38]. In our experiments, the frequency of the “move to” and “contact behavior” were higher in bold males than in bold females. This was particularly notable as the “cheliped display” frequency of shy males was higher than that of bold females, demonstrating the large differences in willingness to fight between males and females. By combining the results of our previous study on fighting between bold and shy swimming crab *P. trituberculatus* [25], we can infer that, in general, categories of fighting willingness and capacity in swimming crabs followed this order: bold male individuals > shy male individuals > bold female individuals > shy female individuals. Except for competition between conspecifics, boldness results in different behavioral strategy, and the combination of individuals with different boldness plays an important role in stabilization and expansion of the population [39]. For example, the bold male squirrels (*Sciurus vulgaris*) rapidly expand the population distribution, and female individuals fully exploit the resources in the known areas [40]. Boldness and other personality traits have broad application in the construction of marine ranches and wildlife protection [41]. Further study and specific application concerning personality need to be studied in the future.

Several studies have shown that the behavior performance of animals is closely related to their energy state and metabolic rate [42,43]. Glycogen and glucose, representing the total energy pool, reflect an individual’s physiological state and are an intrinsic factor affecting behavior performance [44,45]. Previous studies on crustacean species have shown that tissue glycogen concentrations may play an important role in determining personality traits [46]. However, there was no significant difference in the concentration of glycogen sampled in the claw among *P. trituberculatus* with different gender or boldness in our experiment, which was consistent with our previous work [25]. One explanation for this could be that, while bold individuals may have a higher feeding efficiency in the field [47,48], in our experiments crabs were fed by ad libitum at 08:00 every morning with enough clams that all crabs had free access to food, which resulted in similar tissue glycogen concentrations among different types of individuals. As an important metabolic parameter, OCR is also related to behavior performance [49]. In this experiment, the OCR of males was significantly higher than that of females, and the higher metabolic rate likely provided the basis for males’ higher willingness to fight. These results were consistent with those from previous studies, e.g., in fights among juvenile brown trout (*Salmo trutta* L.), individuals with higher OCR had higher social statuses and willingness to fight [50].

Glucose in the hemolymph, as a directly transferable energy, is the primary fuel for ATP (Adenosine Triphosphate) production in crustaceans and is directly related to an individual’s willingness to fight [13]. Hermit crabs (*P. bernhardus*) with higher glucose concentration in hemolymph were more aggressive, with a larger possibility of initiating and winning fights [8]. In the present study, the durations and intensity of fights between bold individuals were higher than those between shy individuals, and the glucose concentration in the hemolymph of bold individuals was lower than that of shy individuals, which may be related to the higher energy consumption in bold male individuals. Furthermore, in addition to the direct consumption of glucose, the mobilization of glycogen, a reserve energy store, also affects the concentration of glucose in the hemolymph [12,51]. In this experiment, the glycogen concentration of the four types of individuals was not significantly different and remained at high levels, indicating that there were no remarkable differences in glycolysis activity. The results of our previous experiments had already shown that bold *P. trituberculatus* have higher movement activity than shy individuals [25]. In a safe environment, bold individuals with higher activity levels will lead to increased glucose consumption, but not increased glycogenolysis [52], which may explain the lower glucose content in the hemolymph of bold males in this study.

Previous studies have indicated that the balance between stored energy in the form of glycogen and mobilized energy in the form of circulating glucose is a key factor determining fighting success [8]. More specifically, it has been shown that the metabolic consequences of fighting include elevated oxygen consumption and energy depletion [11], which has been observed in the shore crab *Carcinus maenas* after fighting [14,15]. The decrease in glycogen and the increase in glucose observed after fighting is consistent with the results from earlier studies of swimming crabs [11,25]. Thus, it appears clear that in response to fighting the glucose concentration is elevated, presumably via the mobilization of glycogen reserves, which could have severe ramifications as crabs incur significant anaerobic debts (accumulation of lactate in the blood) and loss of energy stores proportionate to the duration of fights [14].

Lactate accumulation in the hemolymph can be used as an effective indicator to compare anaerobic respiration among crustaceans [53]. The increased hemolymph lactate contents in the swimming crab after fighting indicated that the intensity of anaerobic respiration was higher during fighting, which was consistent with previous observations in shore crabs *C. maenas* [15]. The significant difference of hemolymph lactate before and after fighting showed that in order to secure energy supply in fighting, high density anaerobic respiration was used by both male and female crabs, which was similar to our previous studies [10,25]. However, there was no significant difference in the concentration of lactate in hemolymph among the four types of individuals after fighting, indicating that the intensity of anaerobic respiration did not differ between females and males during fighting.

## 5. Conclusions

Boldness affected the agonistic behaviors of female and male swimming crabs. The fighting willingness and ability were higher in male crabs than in female crabs and higher in bold individuals compared to shy individuals. Energy reserves and metabolic rates may be critical factors affecting personalities and the associated agonistic behavior in crabs. Our result is helpful to crab culture as well as in enhancement and releasing in marine ranches construction. However, personality is not only related to its energy metabolism, but also affected by genetic inheritance, environmental conditions, and other factors. Therefore, it will be a long-term challenge to further explore the personality characteristics and agonistic behaviors of swimming crabs.

## Figures and Tables

**Figure 1 animals-12-02363-f001:**
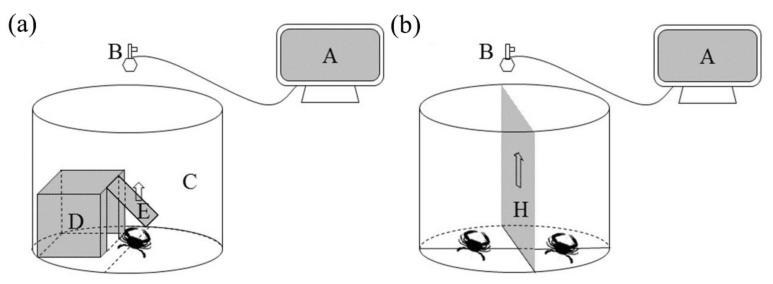
Personality recording system (**a**) and agonistic behavior recording system (**b**). Note: (A) monitor; (B) infrared camera; (C) cylindrical experimental aquarium; (D) box shelter (30 × 20 × 20 cm); (E) trapdoor (20 × 20 cm); and (H) separator board (60 × 50 cm).

**Figure 2 animals-12-02363-f002:**
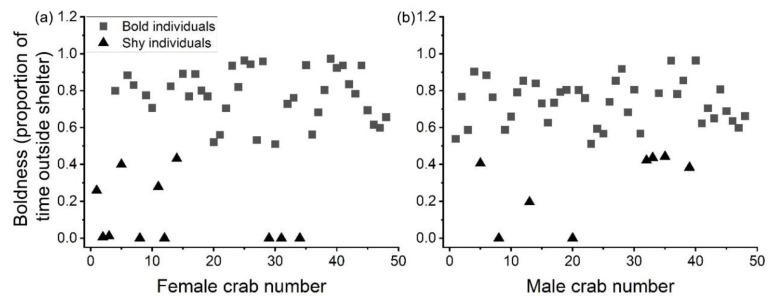
The result of PAML according to the boldness of female swimming crab (**a**) and male swimming crab (**b**).

**Figure 3 animals-12-02363-f003:**
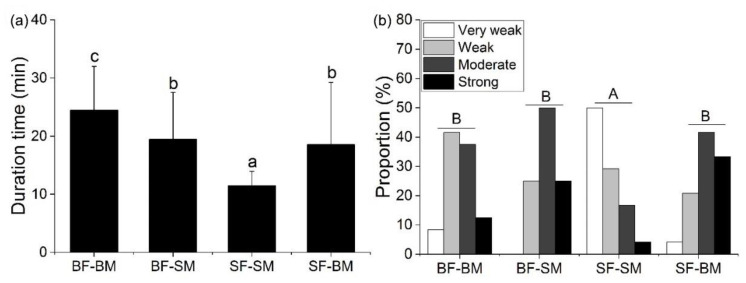
Fighting duration time staged between groups (**a**). The percentage of intensity of fights between groups (**b**). Note: Different lowercase letters represent significant differences between groups (*p* < 0.05); capital letters represent significant differences of intensity between groups (*p* < 0.05).

**Figure 4 animals-12-02363-f004:**
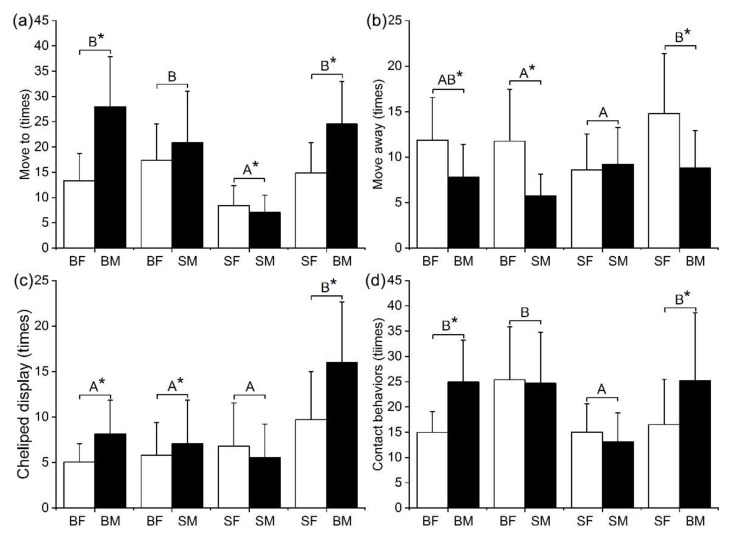
Frequency of agonistic behavior among different groups: move to (**a**); move away (**b**); cheliped display (**c**); and contact behaviors (**d**) of crabs in the fight. Note: Capital letters represent significant difference of agonistic behavior among different groups; an asterisk (*) denotes significant difference of behaviors between two matched crabs. (*p* < 0.05).

**Figure 5 animals-12-02363-f005:**
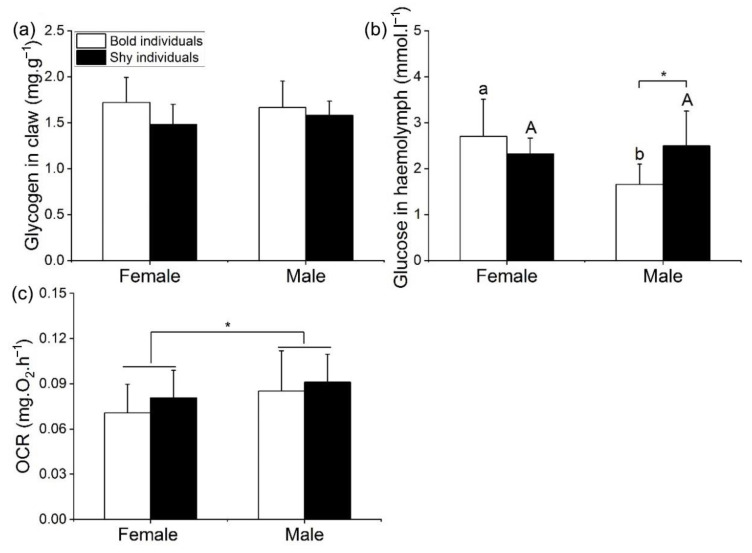
Concentrations of glycogen (**a**) in the claw, glucose in the hemolymph (**b**), and OCR of different individuals (**c**). Note: Capital letters represent significant difference of shy individuals between females and males; lowercase letters represent significant difference of bold individuals between females and males; an asterisk (*) denotes a significant difference of behaviors between two matched individuals. (*p* < 0.05).

**Figure 6 animals-12-02363-f006:**
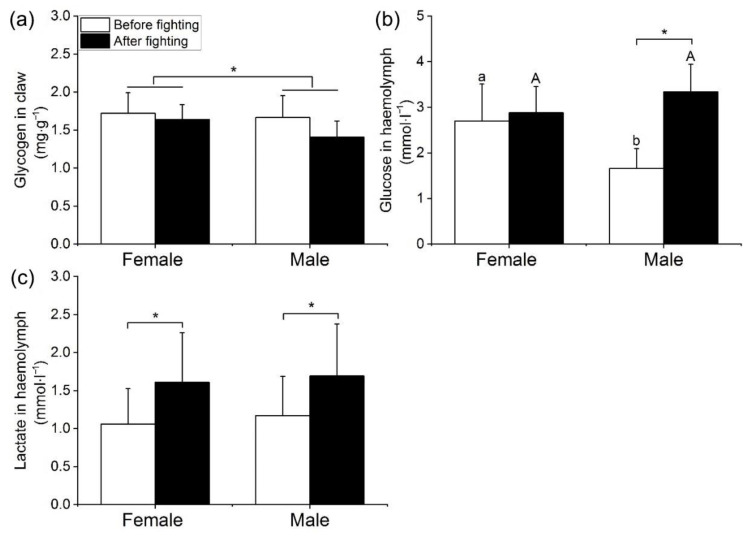
The mean concentrations of glycogen (**a**) in claw, glucose (**b**) and lactate (**c**) in the hemolymph before and after fighting. Note: Lowercase letters represent significant difference of concentrations between females and males before fighting; uppercase letters represent significant difference of concentrations between females and males after fighting; an asterisk (*) denotes denote a significant difference before and after fighting (*p* < 0.05).

**Table 1 animals-12-02363-t001:** Fighting intensity description of swimming crabs.

Intensity of Fighting	Description
Very weak	No physical contact, one crab approached the other and behaved aggressively while the other behaved submissively.
Weak	Both crabs behaved aggressively and grasping occurred. The contest continued until one crab, the winner, successfully climbed on the loser, or the loser repeatedly retreated from the winner.
Moderate	Both crabs entered into a pushing contest until the loser retreated while still holding its chelipeds out.
Strong	Either one crab retreated, i.e., the loser, or both crabs re-engaged in the pushing contest).

**Table 2 animals-12-02363-t002:** Agonistic behavior description of swimming crabs.

Agonistic Behavior	Description
Move to	One crab approaches the other crab
Move away	One crab moves away directly from the crab in an aggressive interaction
Chelipeds display	Chelipeds are held out in front with the chelae open or closed
Strike	One crab suddenly hits out at the other with one or both chelipeds
Grasp	One crab uses its chelae to pinch the carapace, chelipeds, or legs of the other crab
Push	The crab pushes its opponent using its chelipeds, pushing forward with the walking legs and rubs the body of its opponent or grasps using the chelae
Climb on	One crab attempts to or climbs on top of the other crab.
Contact	Consists of “strike”, “grasp”, “push” and “climb on”

**Table 3 animals-12-02363-t003:** Total number of crabs initiating and winning as well as initiating but losing in fights.

Source	BF–BM			BF–SM			SF–SM			SF–BM		
	Init	Init/win	Init/lose	Init	Init/win	Init/lose	Init	Init/win	Init/lose	Init	Init/win	Init/lose
Females	11 ^a^	7 ^a^	4 ^a^	9 ^a^	2 ^a^	7 ^a^	12 ^a^	7 ^a^	5 ^a^	7 ^a^	2 ^a^	5 ^a^
Males	13 ^a^	8 ^a^	5 ^a^	15 ^a^	9 ^b^	6 ^a^	12 ^a^	8 ^a^	4 ^a^	17 ^b^	10 ^b^	7 ^a^

Note: Init, the number of fights initiated by individuals during the fighting; Init/win, the number of wins in fights initiated by individuals; Init/lose, the number of losses in fights initiated by individuals; values with different lowercase letters in the same row are significantly different from each other (*p* < 0.05).

**Table 4 animals-12-02363-t004:** Results of two-way ANOVAs to determine the effects of boldness and sex on the concentrations of glycogen in claw, glucose in the hemolymph, and OCR of swimming crabs P. *tritubercuatus*.

**Source**	**Glycogen in Claw**			**Glucose in Hemolymph**			OCR		
	**MS**	** *F* **	** *p* **	**MS**	** *F* **	** *p* **	**MS**	** *F* **	** *p* **
B	0.189	2.621	0.113	0.133	0.354	0.558	0.001	1.690	0.197
S	0.003	0.046	0.931	1.697	4.508	0.045 *	0.002	4.041	0.047*
B×S	0.042	0.584	0.449	1.739	4.621	0.043 *	0.001	0.112	0.739

Notes: “B” denotes the behavioral type (bold vs. shy) and “S” denotes the sex (male vs. female), and the “B × S” denotes the interaction between behavioral type and sex; an asterisk (*) denotes indicated significant difference in behavioral type and sex. (*p* < 0.05).

**Table 5 animals-12-02363-t005:** Results of two-way ANOVAs to determine the effects of sex and fighting on the concentrations of glycogen in claw, glucose, and lactate in the hemolymph of swimming crabs *P. trituberculatus*.

Source	Glycogen in Claw			Glucose in Hemolymph			Lactate in Hemolymph		
	MS	*F*	*p*	MS	*F*	*p*	MS	*F*	*p*
T	0.509	8.357	0.005 *	5.968	16.590	0.001 *	4.800	13.683	0.001 *
S	0.376	6.175	0.015 *	0.591	1.643	0.212	0.159	0.452	0.504
T×S	0.145	2.377	0.128	3.903	10.849	0.003 *	0.003	0.010	0.922

Notes: “T” denotes the fighting (before versus after fighting), and “S” denotes the sex of the crabs (male versus female), and the “T × S” denotes the interaction between fighting and sex; an asterisk (*) denotes significant difference behavioral type and sex. (*p* < 0.05).

## Data Availability

The data presented in this study are available in the article. Further information is available upon request from the corresponding author.

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
