# Peer review of "Differences in Agonistic Behavior and Energy Metabolism between Male and Female Swimming Crab Portunus trituberculatus Based on the Analysis of Boldness"

_animals, 2022, doi:10.3390/ani12182363_

Round 1
Reviewer 1 Report
Major comments
My main criticism of the manuscript is the lack of novelty. So I suggest that the author should highlight/the impact/benefits results of this study. This manuscript focus on agonistic behavior and energy metabolism between different sex of swimming crab. So, the author should highlight which sector can benefit from this study in the future (e.g., the aquaculture industry can use this kind of results or data to prevent competition during the culture stage or for fisheries management purposes, etc.)
The Materials and Methods section is concise and straightforward. The authors divided the swimming crab into four types of individuals in this experiment: bold females (BF), shy females (SF), bold males (BM), and shy males (SM) using the personality analysis method based on machine learning (PAML). However, the used term for the four “types” in the whole manuscript does not consist. Sometimes the authors used “group”. Please recheck the term used in the whole manuscript again. The results are reported clearly and concisely in the Results section. I did make some suggestions (see minor comments section below). The authors discuss the results fairly well in the Discussion section though I did have some questions regarding some areas that could be improved (see minor comments section below).
Minor comments
Line 43: Keywords: Portunus trituberculatus; Agonistic behavior; Boldness; Sex; Energy metabolism similar to the title. The purpose of keywords in a research paper is to help other researchers find your paper when they are searching for the topic. therefore, putting keywords similar to the title of the study can reduce the search rate for your study. I suggest putting other keywords such as:
Portunidae; crab behavior; machine learning, fighting intensity, crab personality
Introduction
Line 45: “Agonistic behavior, an important form of internal competition, is a fundamental as-pect of ecological and behavioral theory[1]”. This sentence seems not complete. This agonistic behavior for animals or only crustaceans or crabs. Please revised this sentence.
Line 53: Similar to Line 45, this sentence also seems uncompleted. “Oxygen consumption is directly utilized in energy metabolism[9,10]” for animals or only specific species? References 9 refer to invertebrates and references 10 refer to Indian Major Carps. Both references are different species and the study species focused on swimming crab. please explain in more detail for each sentence to make it easier for the reader to better understand.
Line 61-82: It seems that the term used in this Line makes the readers confuses if they did not refer to the references. From Line 61-67, I believe the authors explain the behavior differently for general animal species including land and aquatic animals. Then in Line 67-71, the authors explain the behavior of aquatic animals. Then again, in Line 73, the authors used the word “Animals” in the sentence that is aquatic animals (if we refer to references 28: Yang et al., 2020). Please check the correct term used in each sentence because it can make some confused for the reader. I believe the behavior (either bold or shy types) is different for each species.
Line 74-75: This statement is from the current study result or another research study? If from others research studies, please include the reference. If not, I suggest removing this sentence to the results or discussion part.
Line 83-84: Use the full genus name for the 1st species mentioned here. “P. trituberculatus” change to “Portunus trituberculatus”.
Line 95: I miss the impact of the study. The current version in the Line 95 is weak: “This data will provide a reference for the personality characteristics.” Were there any hypotheses tested?
Materials and Methods
Line 103: What means 12L : 12D? please write in full form.
Line 119: Please describe how to determine healthy female and male crab
Line 144-147: Is there any publications that can support or prove that added food extract can promote the intensity of fighting among the crustacean or crabs? How about if not added the food extract? Is the result will be affected or change?
Line 153-154: please explain and describe the intensity of fighting based on a study by Sneddon et al., 1996 and Matheson and Gagnon, 2012 more detail in the manuscript. I suggest authors include the summary table or figure to explain the intense fighting and agonistic behavior. Please refer to the publication I include below:
Carter, C. G., Westbury, H., Crear, B., Simon, C., & Thomas, C. (2014). Agonistic behaviour in juvenile southern rock lobster, Jasus edwardsii (Decapoda, Palinuridae): implications for developing aquaculture. ZooKeys, (457), 323.
Su, X., Sun, Y., Liu, D., Wang, F., Liu, J., & Zhu, B. (2019). Agonistic behaviour and energy metabolism of bold and shy swimming crabs Portunus trituberculatus. Journal of Experimental Biology, 222(3), jeb188706.
Results
Line 216-217: “The boldness (proportion of time outside refuge)” changes to “The boldness (proportion of time outside shelter)”
Line 221: Figure 2. put the unit for boldness on the vertical axis. example: Boldness (proportion of time outside shelter).
Line 221: Change the description for Figure 2 to Figure 2. The result of PAML according to the boldness of female swimming crab (A) and male swimming crab (B)
Line 236: the description of Figure 3 should be placed after or below the figure.
Line 251-252: Change “SF-SM groups (Figure 4B);” to “SF-SM groups (Figure 4B).”
Line 252-253: “the total number of “cheliped display” of crabs reached the maximum which was significantly higher than that of the other (Figure 4C). How can authors identify the total number of “cheliped display” reach the maximum value? Is the word maximum suitable for this sentence? Maybe higher is more suitable.
Line 255-257: Is that a significant difference for SF-SM group in Figure 4A? It seems no significant difference. Please check. Change the P= 0.046 to two decimal points. Similar to other values in the text.
Line 250-266: Authors explain the result for Figure 4A-C. Where is the result for Figure 4D? Please include the explanation in Figure 4D.
Line 268: Change the description for Figure 4 to Figure 4. Frequency of agonistic behavior among different groups: move to (A), move away (B), cheliped display (C), and contact behaviors (D) of crabs in the fight.
Line 289-291: the description of Table 2 is not complete. Change to: Table 2. Results of two-way ANOVAs to determine the effects of boldness and sex on the concentrations of glycogen in claw, glucose in the hemolymph, and OCR of swimming crabs P. tritubercuatus
Line 293: I cannot find any result about “T × Z” in Table 2. Please check again.
Line 295: the description of Figure 5 should be placed after or below the figure.
Discussion
Line 347-350: This is one of the important results in this study, however, there is still a lack of discussion about why the male crabs (either bold or shy) have higher fighting willingness and capacity compared to female crabs. I believe that the aggressive behavior of certain marine species may be due to territory, protection, and genetic variation. Please add one more paragraph to discuss in detail the reason why this behavior performance happens between males and females of swimming crab.
Line 357: What mean by different categories? Different sex or concentration of glycogen before or after fighting? Results in Figure 5A showed no significant difference but figure 6A was a significant difference.
Line 369: Please write in full form for ATP.
Line 388-390: This statement is opposite to your results in Line 381-383: “Higher activity levels will lead to increased glucose consumption[23], which may explain the lower glucose content in the hemolymph of bold males in this study”. after the fighting is considered as a higher activity? but why the glucose content in the hemolymph is higher for male crabs after fighting in Figure 6B?
Line 395-402: It seems no discussion in this part. Only the authors explain again the results in Figure 6C. Please discuss more why Lactate in the hemolymph is higher after fighting for both male and female
Conclusions
Lines 404–407: Lack of important information in the results for the conclusion. The author should highlight the impact/benefits results of this study
Reviewer 2 Report
A well-planned and explained study
Generally, a very well written paper but it needs proof-reading.
· Line 106: were (not was)
· Line 147: evenly
· Line 157: contacting
· Line 158: first
· Line 159: caused its opponent to retreat
· Line 185: were removed
· Line 186: the claw
· Line 224: varied (past tense)
· Line 318: I would say “An asterisk (*) denotes” (as written this is difficult to see)
· References 460, 475, 480: Capitalize the only first word in a title, as is the format used in this journal. (Some journals ask the authors to capitalize the nouns but this journal does not do so).
Reviewer 3 Report
Review of manuscript 1848516 “Differences in agonistic behavior and energy metabolism between male and female swimming crab Portunus trituberculatus based on the analysis of boldness” submitted to MDPI-Animals.
The authors determine whether there are differences in behavior and oxygen consumption and glucose, lactate and glycogen levels among male and female crabs categorized as “bold” or “shy” before and after fights. They conclude that fighting willingness and ability is higher in male crabs than in female crabs. Metabolic parameters varied and are explained by the authors in relation to consumption of energy resources during fighting.
This paper is similar to another paper by the same authors that I reviewed previously for Animals and adds information on males versus females to that earlier study. Again, I am not familiar with this particular area of research so cannot comment specifically on the research but the methods employed and conclusions drawn seem reasonable to me. The subject matter is suitable for “Animals” and I would imagine that this information would be of interest to others working in this area.
For the most part, I will comment only on the manuscript itself. Overall it is well written with only minor grammatical issues. Comments, suggested edits and questions per line:
line 51 – “... the essence of fight is the competition of total energy pool…” is confusing and should be rewritten
57 – add “are” after “and”
75 – change “has competitive” to “have a competitive”
77 – delete “in”
86 – delete “the” before “boldness”
92 – change “Besides” to “In addition”
95 – change “This” to “These”
100 – include the Latin species name after or instead of “swimming crabs”
116 – the use of A and B labels for parts of the apparatus is confusing when A and B represent the two sub-figures / I suggest using lower case letters to indicate the apparatus parts
121 – delete “for”
123 – add “and” after “optionally,”
128 – spelling typo – “chalssification” should be “classification”
141-142 – change “category” to “categories”
143 – reword to “Crabs were marked with a white dot using acrylic paint on the carapace…”
145 – “before the board and crabs were moved into” is confusing and should be rewritten
155 – change “was” to “were” and I suggest changing “referring to” to “according to ” or “as in”
157 – change “contact” to “contacting”
158 – “which win most fighting during recording” is confusing and should be rewritten
159 – reword “retreats its opponent” to “forced its opponent to retreat”
165 – delete “the”
168 – begin new sentence with “In order to…”
169 – You cannot begin a sentence with a numeral / change “3” to “Three” or reword
180 – same as in line 169 / change “8” to “Eight”
182-185 – I suggest rewording to “…using ice [20], and a hypodermic needle with a 1 ml syringe was used to take 2 ml hemolymph samples from the base of the last pereopod [29].” After hemolymph was collected, the carapace and claw were removed …”
186 – insert “the” before “claw”
189 – delete “the”
191-192 – I suggest rewording to “…in the claw tissue (mg g -1) using spectrophotometry at 505 nm.”
193-196 – reword as in 191-192
199 – change “was” to “were”
207 – I suggest rewording as “…ANOVAs with sex and boldness set as the fixed factors.”
209-210 – same wording as in 207
216 – include the species name again here at beginning of Results
218, 219 – delete “the” in both lines
227 – add “both” after “in”
234 – delete “intensity”
239 & 270 – the use of A and B to indicate significance is confusing since A and B refer to the sub-figures
275-276 – do the authors mean “lower-case letters”? / there are no “capital letters”
283 – change “male” to “males”
285 – same comment as for line 283
293 – what does “as well as the T x Z” refer to? I don’t see that in the table
294 – the bold numerals are not very distinctive / I suggest denoting with a * instead
301 – “between before and after fighting” is confusing / just delete “between”
308 – change “of” to “in”
309 – delete “between”
315 – delete “between”
325 – same comment as for line 294
327 – change “individual” to “individuals”
339 – change “have” to “had”
354 – delete “of”
360 – “ad libitum” should be italicized (Latin)
370 – suggest change to “Hermit crabs (P. bernhardus)…”
398 & 402 – delete “the”
435 – Pagurus bernhardus should be italicized
499 – reference page numbers missing
Round 2
Reviewer 1 Report
The authors already revised by following the comments.